# Towards a Knowledge guided Multimodal Foundation Model for Spatio-Temporal Remote Sensing Applications

## Abstract

In recent years, there has been an increased interest in foundation models for geo-science due to the vast amount of Earth observing satellite imagery. Existing remote sensing foundation models make use of the various sources of spectral imagery to create large models pretrained on the task of masked reconstruction. In this paper, we present a foundation model framework, where the pretraining task captures the causal relationship between multiple modalities. Our framework leverages the knowledge guided principles that the spectral imagery captures the impact of the physical drivers on the environmental system, and that the relationship between them is governed by the characteristics of the system. Specifically, our method, called MultiModal Variable Step Forecasting (MM-VSF), uses forecasting of satellite imagery as a pretraining task and is able to capture the causal relationship between spectral imagery and weather. In our evaluation we show that the forecasting of satellite imagery using weather can be used as an effective pretraining task for foundation models. We further show the effectiveness of the embeddings produced by MM-VSF on the downstream tasks of pixel wise crop mapping and missing image prediction of spectral imagery, when compared with embeddings created by models trained in alternative pretraining settings including the traditional single modality input masked reconstruction.

## 1 Introduction

Increased availability and ease of access to large scale satellite data has motivated the development of deep learning models that use this data to perform tasks such as land cover mapping Ghosh et al. (2021b); Kussul et al. (2017), wildfire mapping Nayak et al. (2018); Seydi et al. (2022); Zhao et al. (2018), crop yield prediction Kuwata et al. (2015); You et al. (2017), flood forecasting Bentivoglio et al. (2022) etc. In recent times, methods have been developed to use large amounts of data in a self supervised fashion to pre-train their model weights using a pre-training task, such as masked reconstruction He et al. (2022), which would then be fine-tuned for downstream tasks. Such models, called foundation models, have been shown to perform well (after refinement) over various downstream tasks, in the image Yuan et al. (2021); Singh et al. (2022) and text domains Touvron et al. (2023) (e.g. large language model such as GPT Achiam et al. (2023)).

Motivated by the success of such models, there is a huge interest in building geo-science foundation models for remote sensing applications Mai et al. (2022). Current remote sensing foundation models are typically built using the vast amounts of spectral data available from various satellites Jakubik et al. (2023); Gao et al. (2022); Cha et al. (2023); Mendieta et al. (2023); Liu et al. (2024). These models use a pretraining task such as masked reconstruction Cong et al. (2022); Tseng et al. (2023) where given a sequence of satellite imagery, the objective is to reconstruct the spectral imagery of the masked timestamps in the sequence. Utility of finetuning such models have been demonstrated for downstream tasks such as flood inundation mapping, wildfire scar mapping, cloud removal, urban semantic segmentation mapping, scene classification etc. Jakubik et al. (2023); Cong et al. (2022); Sun et al. (2022).

In this paper, we present a foundation model, where the pretraining task captures the causal relation between two different modalities. To understand the reason behind choosing such a pretraining task, let us look at Figure 1, which represents an environment as a system, where various physical

Figure 2: *Abstract Representation of our proposed Variable Step Forecasting Pretraining task. Our pretraining task is to estimate a spectral image in future(red) using satellite imagery and weather context(yellow) and weather data till that day in future(green)*

drivers such as weather (e.g. temperature, precipitation) act upon a region (e.g. a farm, hydrological catchment) to result in a response such as crop growth, streamflow, greenhouse gas emissions. The relationship between drivers and response is governed by the physical properties of the environmental system (e.g. land cover type, soil characteristics). Some characteristics of the environmental system can be observed by in-situ or remote sensors (e.g., satellites such as Sentinel, Landsat).

Our pretraining task attempts to capture the causal relationship between weather and satellite imagery by predicting future satellite imagery based upon past weather and satellite imagery (see Figure 2). Note that while weather is expected to be available continuously (at daily scale), the satellite imagery may be available only for a small number of dates (at least one). We call this pretraining task "variable-step forecasting". A foundation model trained using such a pretraining task is likely to perform better on downstream

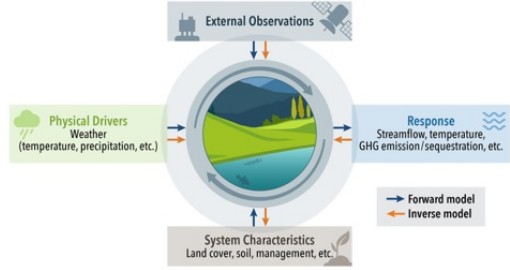

Figure 1: *Abstract Representation of an Environmental System depicting the various components*

tasks, where such a relationship between modalities is relevant. For example, such a foundation model can perform better on the downstream tasks of mapping of crops Ravirathinam et al. (2024) at a pixel level and filling missing or noisy satellite data on a future date using satellite observations from past dates because this model captures the causal relationship between weather drivers and spectral data from the satellite.

Specifically, we present a novel spatio temporal multimodal foundation model framework called MultiModal Variable Step Forecasting (MM-VSF), that captures the casual relationship between satellite and weather data (two entirely different modalities) to perform the pretraining task of forecasting (variable step forecasting). Our architecture is temporal flexible (i.e., its input does not need to be of a certain length for it to make prediction), which is useful for generalizing across a wide range of downstream tasks. Via experimental evaluation on two different downstream tasks of crop mapping and corrupted/missing image prediction, we show that the embeddings created by our framework are richer when compared to those created using the pre-training task of reconstruction.

Our key contributions are listed below:

- We propose MM-VSF, a temporally flexible foundation model framework that captures causal relationships across modalities .
- To help capture causal relationships we propose variable step satellite imagery forecasting with multimodal data as a novel pretraining task for geoscience remote sensing models.
- Our pretraining task is guided by the knowledge that the spectral imagery captures the impact of the physical drivers on the environmental system.
- We demonstrate how embeddings from MM-VSF can be efficiently fine-tuned for spatiotemporal downstream tasks, showing improved results in crop mapping and missing image imputation relative to foundation models built using masked image reconstruction.
- We release the code and model used to public (Link)

## 2 RELATED WORK

There are a number of existing geoscience foundation modelsJakubik et al. (2023); Cha et al. (2023); Mall et al. (2023); Deng et al. (2024); Hong et al. (2024); Guo et al. (2024); Mai et al. (2022); Bastani et al. (2023). Most of these foundation models can be placed into one of two groups based on the data they use : (1) weather-climate Nguyen et al. (2023); Pathak et al. (2022) that are typically used for weather forecasting or climate modeling and (2) spectral data from remote sensing satellites Cong et al. (2022); Jakubik et al. (2023) that are largely used for identifying land-use land-cover change dynamics. The most common pretraining task in geoscience has been reconstruction of spectral imagery. To enrich the embedding created, varying amounts of the input spectral image are masked, making the reconstruction task of the entire image harder leading to better embeddings Cong et al. (2022). However, simple reconstruction embeddings capture just that particular image and might not be suited for downstream tasks that rely on multi temporal contexts such as crop mapping or land cover land use change. To solve this, previous works included multiple timestamps in their input, however some of these methods stacked these images together Jakubik et al. (2023), thus removing the temporal aspect. However, some methods added a timestamp positional embedding so that the model has a sense of time Khanna et al. (2023); Cong et al. (2022). This led to moderate success in handling downstream tasks that require multi temporal contexts. Another common pretraining task is forecasting of imagery. Typically, this pretraining task has been used in weather related foundation models and not in spectral imagery based foundation models. Foundation models created using masked forecasting has shown great success in weather related downstream tasks Schmude et al. (2024); Pathak et al. (2022). To enrich the embeddings created, these works add variable future time forecasting, i.e vary the amount of time into the future the model needs to forecast, which was achieved by including an embedding for delta time Nguyen et al. (2023). Other variants of foundation models include diffusion models that incorporate more information such as geographic location, time of year, country etc Khanna et al. (2023).

## 3 ARCHITECTURE

Our architecture follows a heavy encoder and lightweight decoder format. Keeping our decoder lightweight forces richer embeddings from encoder, suitable for pretraining and downstream tasks. Unlike previous models, we incorporate multiple modalities (spectral imagery and weather) in pre-training our architecture. Figure 3 shows MM-VSF's architecture. For the forecasting-based pre-training task, it is essential for architecture design to capture spatial and temporal modalities of satellite data, temporal weather data, and their interactions.

### 3.1 SATELLITE IMAGE ENCODER/DECODER

We use a shared Vision Transformer (ViT) to extract spatial features from spectral imagery across timestamps. ViT have been shown to be effective in the presence of high masking He et al. (2022), even in geoscience contexts Jakubik et al. (2023). The ViT converts each image into a patch grid of embeddings, incorporating patch positional information. This results in a series of spectral image embeddings on unmasked patches for each timestamp. Since our input is an image series, we propose using a shared ViT across timestamps, leading to a robust encoder, that is capable of embedding images from all timestamps.

### 3.2 WEATHER ENCODER

Due to the coarse spatial resolution of weather data, typically for each image location, we have one weather data point value per timestamp. Thus we use a sequence-to-sequence Bidirectional LSTM (BiLSTM) to encode the weather data. The Bi-LSTM based approach showed higher accuracy over the transformer based approaches when trying to solely reconstruct masked weather. The BiLSTM generates weather embeddings for each timestamp, which are then subsampled to match input image timestamps, similar to WSTATT Ravirathinam et al. (2024), called temporal embedding matching.

### 3.3 TIMESTAMP/DELTA ENCODER

For temporal information we incorporate day of year (DOY) using a shared linear layer with tanh activation, creating DOY embeddings for each timestamp. Additionally, we create another series that corresponds the number of days in between the images, i.e the delta in timestamps. We generate embeddings for these time delta between images using a separate linear layer. The DOY embeddings

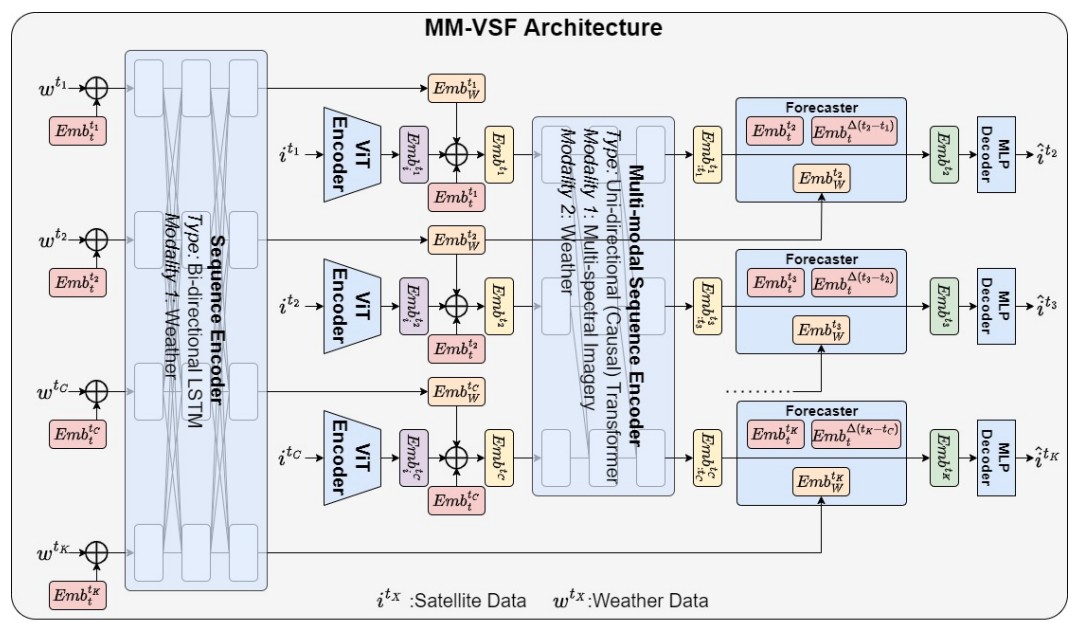

Figure 3: *MultiModal Variable step Forecasting (MM-VSF) Architecture diagram*

provide temporal context, while delta embeddings, shown useful in forecasting tasks Nguyen et al. (2023), informs the model about forecast distance.

### 3.4 MULTIMODAL SEQUENCE ENCODER

From the previous steps we have spatial, weather, and day-of-year embedding series. Since all these series are of the same length, we add them all along the temporal dimension to create the multimodal embedding series. We use a transformer with forward-only attention to extract the spatio-temporal information from this multimodal embedding series, analyzing embedding patches from the same spatial location across timestamps. Further we add forward only attention (causal) in transformer, i.e the temporal embeddings created are not bidirectional in nature. This feature makes sure there is no information leakage from future timestamp information to previous timestamps embeddings. The resulting embedding series $Emb_{STW} = [Emb_{:t_1}^{t_1}, \ldots, Emb_{:t_C}^{t_C}]$ can be used for downstream tasks, with flexibility in embedding selection. Depending on the task, one can choose to use all the embeddings or choose to use only the final embedding.

### 3.5 FORECASTER

Since our embeddings are constructed using forward-only attention, we use each of the embeddings in $Emb_{STW}$ to forecast an image in its respective future. For each embedding $Emb_{:t_i}^{t_i}$, the forecaster uses the weather embeddings $Emb_w^{t_j}$ as well as the the temporal embeddings $Emb_t^{t_j}$ and the delta embeddings to generate the embeddings $Emb^{t_j}$ of the forecast timestamp $t_j$. By incorporating weather data leading up to a particular date, we can make more informed estimates about how the land cover might appear, as weather patterns play a significant role in shaping the landscape over time. We combine the four embeddings through addition and feed it to a series of linear and activation layers that forms our forecaster. The objective of these layers is to morph the embeddings from current timestamp to the future timestamp before passing them to the MLP decoder to get the satellite image forecast.

For all but last embeddings in the series $Emb_{STW}$, we forecast the next time-step, i.e t1 of the input series would be used to forecast the image at t2, t2 would be used to forecast t3 and so on. For the last embedding $Emb_{:t_C}^{t_C}$ we forecast the K'th time-step, which is K-C days into the future. This K is a variable and is sampled for each training instance in a batch, thus the name variable-step forecasting (VSF). This method enables the model to predict future images based on current image embeddings and weather information up to the forecast date, acknowledging weather's impact on land cover. Moreover, such multiple forecasting allows for a robust decoder, as similar to encoder, the decoder also has shared weights across timestamps.

### 3.6 DECODER

Following this stage, we repopulate these embeddings in their respective unmasked positions in the timestamps and zero out the masked patches to pass to the decoder, as done by most methods in transformer based autoencoder methodsJakubik et al. (2023); He et al. (2022). We use a light-weight MLP decoder (similar to the one used in ViT) to ensure that the main focus of the model is to create strong encoder to capture the best information so that even a lightweight decoder can perform the pretraining task required. Similar, to the encoder, we have a shared decoder that performs the operations on each timestamp using the same weights. The decoder maps the embeddings to the spectral image space and reshape the output to match the required size of the spatiotemporal stack.

## 4 PRETRAINING

As mentioned before, our proposed pretraining task is forecasting of satellite imagery using multimodal data in a knowledge informed fashion, a pretraining task that varies significantly from the traditional single modality reconstruction. Our proposed approach can be trained using the architecture described above in a direct shot, i.e initialise and update all layers at once. However, this may not be the best way to ensure that the $Emb_{STW}$ embeddings would contain information about the relationship between modalities, as that is the ultimate aim of our proposed approach. To ensure this information capture, we propose a phase wise pretraining process. Specifically we propose 2 phases of pretraining, where the first phase focuses on getting information from each modality and the second phase focuses on encoding the relationship between the modalities in the embedding.

**Phase 1: Masked Reconstruction** As can be observed, there are many components to our architecture. However, if you break it down, each component serves a primary purpose. The VIT encoder/decoder serves the purpose of bringing an image to and out of embedding space. This component can be separately trained via masked reconstruction of satellite imagery. Similarly, the Weather encoder brings the weather data to embedding space. This encoder can also be constructed via a masked autoencoder approach using a Bidirectional LSTM. These steps result in models that give embeddings for both the spectral imagery and weather components. We can now add the multimodal sequence encoder and train it using the spectral imagery model with the overall objective to reconstruct the entire image series, but instead of masking images, we can mask embeddings, to make the sequence encoder stronger. After these steps, we have a baseline for the $Emb_{STW}$ embedding series(through addition). Though this embedding series contains spatial and multitemporal information about the input series, it would not capture the relationship between modalities. To prove this, this embedding series would be used a baseline for finetuning across our downstream tasks. To infuse multimodal relationship, we move to Phase 2, adding a forecaster.

**Phase 2: Forecasting** From the architecture section, we saw that the forecaster was a set of linear layers, making it sound simple. However its purpose is to translate the embedding from input timestamp space to the forecasted timestamp space. Now, to perform this translation within a few layers is very difficult, so the model will search for ways to learn this relationship from other parts of the overall architecture. The solution would come from the multimodal data, in our case weather data. Using weather data, the current embedding, timestamp embedding and delta embedding, the forecaster is tasked to create the future embedding, but due to less amount of layers, the embeddings of all the components will all change to best capture all information from each modality in regard to forecasting, thus capturing the relationship across modalities. For example, if there was a lot of rain the lakes would be fuller, if there was a lot of sunlight then growth of crop/vegetation would accelerate. As a result, this scheme would infuse land growth/change dynamics into the embeddings that the model creates. Asking the model to do this without weather data would be very challenging as it would have no relationship to learn, tasking the forecaster with a big challenge. This no weather variant of embeddings is also added a baseline for downstream tasks.

To summarise, our proposed pretraining task is to predict a spectral image in the future (response) using a series of spectral images in the past (context) along with the weather till the future date (query) we want to predict. We call this pretraining task as Variable step Forecasting ($VSF$), where the model is expected to forecast k steps into the future. To ensure that relationship across modaliteis is captured in the embeddings, we adopt a two phase pretraining process. Our hypothesis is that this extra knowledge infusion would help greatly in downstream tasks that rely on land growth and change dynamics such as crop prediction, land cover land use change, etc. A schematic of the stages of pretraining can be found in appendix A.1. We mask out patches in the spectral imagery and

weather data in all the stages to make the prediction harder and lead to better embeddings. We will describe our masking strategy for pretraining in a future section(Sec 5.2).

# 5 DATASET

## 5.1 DATA SOURCES

Our spectral imagery data comes from Sentinel imagery Drusch et al. (2012) and our weather data is from ERA5 land analysis data Hersbach et al. (2020). We chose these two sources due to their temporal resolutions and more importantly their availability globally from 2021. Due to these factors, we randomly sampled around 10000 locations across land areas globally. Each location is of size 128x128 Sentinel pixels and for spectral imagery we collected all images for that region in that year. Due to missing data and improper coverage of some regions, the number of Sentinel samples from each region would vary. For example, regions in well covered regions such as US, might have upto 70 image instances in a year for that region, whereas regions like India (which is not as well covered) would have 40 image instances. For each instance, we collect six bands namely (B2, B3, B4, B8, B9, B12), which have shown to be the most useful in land cover related tasks and have been used in other works Jakubik et al. (2023). For each image instance, we also collect the day of the year it came from, thus forming a series with values from 1 to 365 and a length the same as the number of image instances for that location. Though ERA5 data source consists of various bands that are useful for land cover related tasks we chose 5 bands to keep it simple and efficient, namely (temperature 2m min, temperature 2m max, total precipitation sum, $u$ component of wind 10m, $v$ component of wind 10m). ERA5 data is available at a daily temporal resolution and a spatial resolution of 11k meters. Given the coarse spatial resolution compared to Sentinel, for most locations in our analysis we get only one value band set per timestamp, thus making our weather data of length 365 with 5 values per timestamp.

To summarise, for each location our data comprises of 3 main components:
• **Spectral Imagery Series**: A series of Sentinel2 Imagery each of 6 bands and of shape 128x128. Length of this series depends on coverage of the location.
• **Weather Data Series**: A series of ERA5 Land data of 5 bands and of shape 1x1, with a series length of 365 (one per day)
• **Day of Year Series**: A series of the day of the year number for each spectral image in the series. The length of this series is same as the spectral imagery series.

## 5.2 MASKING

Since we are dealing with a spatiotemporal input and architecture, we adopt a spatiotemporally uniform masking method, i.e, masking that is fair both spatially and temporally. In our unique masking strategy, there are an equal number of masked patches per timestamp as well as an equal number of masked patches per spatial patch location along the temporal axis.

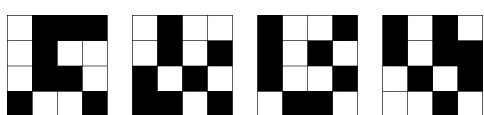

Figure 4: *Example of 50 percent spatiotemporally uniform masking on a 4x4 4 image timeseries*

Figure 4 shows an example of such masking for a image series of 4 4x4 grids with 50% masking. From the Figure, we can see that in each timestamp image there are 8 patches masked and focusing on a particular patch location along the temporal dimension we notice that 2 patches are available. This ensures that all temporal patch series that can be created from unmasked patches at each spatial location would be of same length, easing the implementation of temporal components. Also the shared vision transformer will also have same number of outputs per timestamp due to same number of unmasked patches per timestamp.

# 6 EXPERIMENTAL EVALUATION

## 6.1 BASELINES

Recall that our model uses the multimodal (MM) satellite and weather data as input series and variable-step forecasting (VSF) as pretraining task. In the experiments, we evaluate the effectiveness of MM-VSF by comparing with several representative foundation modeling methods. We create these baselines by varying the choices for input series and pretraining task, as described below.

- **SM-MAE**: Single Modality Masked AutoEncoder (SM-MAE) with satellite data as input series and masked reconstruction as pretraining task.
- **MM-MAE**: MultiModal Masked AutoEncoder (MM-MAE) with satellite and weather data as input series and masked reconstruction as pretraining task
- **SM-VSF**: Single Modality Variable step Forecasting (SM-VSF) with the satellite data as input series and variable-step forecasting as pretraining task.

Note that the above combinations can be implemented by only changing the inputs and the loss functions, without significant architecture changes. Thus these variations, which we call baselines, can also be considered as ablations of different components of our model. Though MM-MAE uses multi modal data we hypothesise that due to the task of reconstruction the relationship between modalities will not be captured in the embeddings. Also note that SM-MAE is closest to the existing remote sensing based foundation models (e.g.,Jakubik et al. (2023); Cong et al. (2022); Sun et al. (2022)), as these works also use only satellite data to pretrain using Masked AutoEncoder(MAE).

### 6.2 Implementation Details

For the pretraining phase, we choose an input series length of 6 images, and selected a random image after the final image as the final image to forecast. Since our input series length is 6, we can create multiple samples from one location, i.e one location's data can result in multiple input series. After splitting the 10000 sampled locations in 60-20-20 split and created numerous samples in each split, We used 50000 image series for training, 10000 images series for validation, and 10000 for testing. We also use 50% spatiotemporally uniform masking for both forecasting and reconstruction based pretraining. We use a patch size of 8 for the vision transformer and a hidden dimension size of 256. Model was trained on 4 A100 Nvidia GPUs using Adam Optimizer and Mean Squared Error loss.

## 7 Results

In this section we evaluate various aspects of our proposed foundation model framework. First, we evaluate the performance on the forecasting-based pretraining task and highlight some examples to show that our method's embeddings capture aspects that go beyond just encoding the image. We further evaluate the performance of our embeddings against other variants when finetuned to the downstream tasks of crop type mapping and missing spectral imagery imputation.

### 7.1 Results on Pretraining Task

Here we evaluate the relative utility of using both weather and spectral data (MM) vs spectral data (SM) only on the forecasting-based pretraining task. With the help of a few examples, we illustrate that the embeddings produced by MM-VSF are more powerful than those SM-VSF, as MM-VSF is able to capture the dynamic relationship between weather and the changes in the physical environment on the land captured by spectral data. Figure 5 shows a comparison of the images from these models,i.e SM-VSF and MM-VSF, on 3 independent examples. Each row correspond to a sample, with the first 6 images corresponding to the satellite component of the input series to the model, the weather component is not shown in the image but is passed along with the satellite component(as shown in Figure 3). Now, the output would also be a series of 6 images, with the last image corresponding to a future day as specified in the user input (shown above the image in the groundtruth column). Note that Figure 5 only shows the final forecast image for each method, as this is where one would expect the most impact from using weather. Both schemes are able to construct earlier images quite well, although MM-VSF can improve over SM-VSF in most cases.

Row 1 depicts an example of a crop field, with the final forecast image being 120 days following the 6th image in the input series. We can see that from the groundtruth image, that harvest has occurred in the circular fields and growth has happened in the top left corner field. Comparing the forecasted images from SM-VSF and MM-VSF, we can see that MM-VSF is able to capture both the harvest and the growth of the crops whereas SM-VSF is not able to capture these changes. This shows that the inclusion of weather allows the embeddings created by MM-VSF to capture land cover dynamics that are driven by weather. Such dynamics cannot be captured by SM-VSF, as it only has access to the past spectral imageries, and thus captures temporal autocorrelation amongst the spectral images. Row 2 shows another example of a crop field later in the year, with the final forecast image being 90 days in future. We can see from the groundtruth image 90 days later that snow is present in the field, which is captured by MM-VSF but not SM-VSF, whose prediction shows a faded green field. This

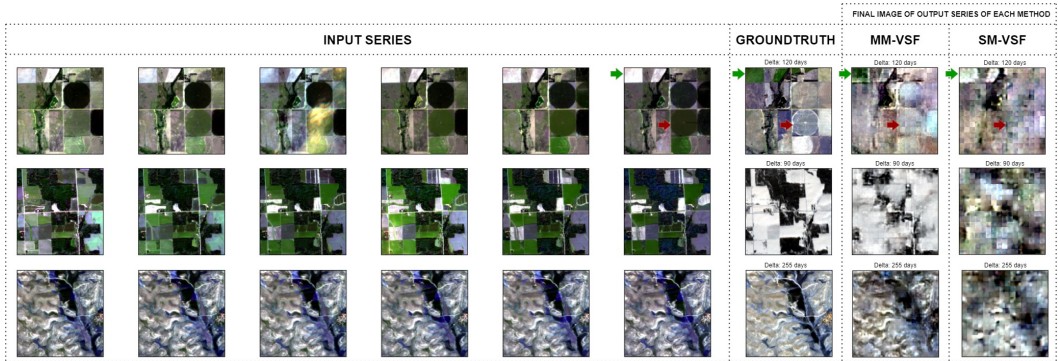

Figure 5: *Forecast based Pretraining Task comparison. 50% Masking is not shown. Row 1 depicts a crop field with, Green arrows depict regions of growth, and Red arrows depict regions of harvest. Note how MM-VSF captures both these phenomena better than SM-VSF. Row 2 depicts an example where MM-VSF is able to add snowfall accurately compared to SM-VSF. Row 3 depicts an example where MM-VSF does not change land cover due to terrain but SM-VSF adds false greenness*

illustrates the ability of MM-VSF to capture the relationship between precipitation and temperature (i.e., precipitation during cold winter days can fall as snow). One can also notice that evergreen regions within the forecasted image of MM-VSF have less snowfall when compared to the fields, showing that terrain information is also being captured. This is further reflected in Row 3, where a mountainous region is depicted and the final forecast image is 255 days in future. We can see that even after 255 days there is not much change in the region, which is correctly captured by our method (MM-VSF) whereas SM-VSF seems to add some false greenness. These examples illustrate that embeddings created with the inclusion of weather contain relevant land cover dynamics and terrain information, which is very useful for various downstream tasks. We further observe that all images from the forecast of SM-VSF appear blocky, inspite of long periods of training. This can be explained as the model unable to forecast accurate images without weather information.

We also evaluated SM-MAE and MM-MAE, and observed that both models reconstruct the image series quite well, with MM-MAE doing slightly better than SM-MAE. For more visuals on this comparison please refer to Appendix A.2

### 7.2 Downstream Task: Crop Mapping

Here we evaluate the performance of our proposed method (MM-VSF) in producing embeddings by fine-tuning to the pixel wise crop type mapping downstream task. We compare our results with embeddings created using other pretraining tasks and modalities.

### 7.2.1 Dataset and Region of Analysis

Our data for finetuning comes from Sentinel2 and ERA5 land data for the region of the T11SKA Sentinel tile in the California Central Valley, a region rich in various crop classes and has been used for crop type mapping in various other worksGhosh et al. (2021b;a); Ravirathinam et al. (2024) from the years 2018 and 2019. Like other works, we get our labels for this region from the Cropland Data Layer(CDL), an annually released land cover map for the entire continuous US by the USDA. A diagrammatic representation of the CDL labels and the geographic location of the T11SKA tile can be seen in Figure 9 in the appendix. Similar to WSTATT Ravirathinam et al. (2024), we adopt a grid based training method, by splitting the entire region into train, validation and test grids and also follow their preprocessing steps including combination and erosion.

### 7.2.2 Crop mapping Architecture and Implementation details

To perform crop mapping we would need an architecture that gives us a pixel wise output, in particular, we would need a decoder that takes the embedding series given by the foundation model encoder, i.e $Emb_{STW}$, using it to construct a pixel wise classification map. To map this embedding series to a pixel wise map, we follow an attention based approach, similar to WSTATTRavirathinam et al. (2024). This strategy assigns a weightage to each timestamp and does an aggregated sum to form a multitemporal attention-based embedding. This multitemporal embedding is then acted on by a series of upscaling and convolution layers with activation, and an output Linear layer to form a pixel wise map. Figure 10 in appendix shows a schematic of the architecture used for

crop mapping. For our input series, we chose to have 10 spectral images, biweekly from May to Sept, and while finetuning for crop mapping there is no masking of spectral imagery done. Note that the number of timestamps passed in the downstream task is different from the number passed during pretraining. This highlights the temporal flexibility of our approach. In all methods(MM-VSF, SM-MAE, etc.) the encoder weights to obtain the embedding series are fixed from the respective pretraining task and the only the attention mechanism and decoder layers are finetuned. All finetuning is done with only 2018 data with respective best hyperparameter settings.

### 7.2.3 PERFORMANCE ON CROP MAPPING TASK

Table 1 compares the performance of MM-VSF and other baselines when finetuned on crop mapping downstream task. Though finetuning is done using 2018 data, testing is done using 2019 data, assessing the robustness of the approaches. We can see that MM-VSF performs better than other variants in almost all classes, showing great improvements in Corn and Walnut. We also observe that MM-

Table 1: *Comparison on downstream task of crop mapping across the pretraining tasks. Finetuning is done only using 2018 data.*

| Crop Class | 2019 Test Classwise F1 Scores | | | |
| --- | --- | --- | --- | --- |
| | SM-MAE | MM-MAE | SM-VSF | MM-VSF |
| Corn | 0.4135 | 0.4717 | 0.4489 | **0.5708** |
| Cotton | 0.8346 | 0.8403 | 0.9055 | **0.9125** |
| WinterWheat | 0.1156 | 0.0770 | 0.1043 | **0.1777** |
| Tomatoes | **0.7680** | 0.7637 | 0.7223 | 0.7341 |
| Grapes | 0.7398 | 0.7502 | 0.7447 | **0.7543** |
| Almonds | 0.3851 | **0.4386** | 0.2073 | 0.2990 |
| Walnut | 0.0238 | 0.1417 | 0.4494 | **0.5384** |
| Pistachio | 0.5070 | 0.6734 | 0.6200 | **0.7003** |
| Alfalfa | 0.6892 | **0.7271** | 0.7072 | 0.7057 |
| Grass | 0.7760 | 0.8436 | 0.7715 | **0.8445** |
| Urban | 0.6111 | **0.6408** | 0.6229 | 0.6191 |
| Average | 0.5331 | 0.5789 | 0.5731 | **0.6233** |

MAE performs better than SM-MAE, which is because inclusion of weather in input is in general better for crop mapping task Ravirathinam et al. (2024), however MM-MAE lacks capturing the relationship between modalities, thus its inferior performance when compared to MM-VSF. Finally, the ability of MM-VSF's embeddings to generalize across years after finetuning indicates that they capture crucial information going beyond what is just present in spectral imagery. Note that with the minimal effort of choosing a timeframe for input series and adding a task-specific decoder, we adapted our MM-VSF framework for the Crop-mapping downstream task providing empirical evidence of our framework's flexibility to adapt to various remote-sensing based spatiotemporal tasks.

### 7.3 DOWNSTREAM TASK: MISSING IMAGE PREDICTION

Here we evaluate the performance of our proposed method (MM-VSF) in the missing image prediction downstream task, i.e estimating the spectral values for missing pixels in imagery. Like the crop mapping task, we compare our results with SM-VSF, MM-MAE, and SM-MAE.

### 7.3.1 DATASET AND PROBLEM SETTING

With the vast amount of satellite imagery being captured on a daily basis, corrupted or missing data is an occurring phenomenon. In some cases, the areas of missing/corrupted data is provided as a mask, but in other cases this mask is not provided, making it hard clean or filter data, thus leaving researchers to use these corrupted images in their work. In this downstream task, we aim to fill these corrupted/missing values in the absence of these masks, i.e the methods have no access/information as to which pixels are corrupted, making it a very real world downstream task. To solve this task, we aim to use the spatiotemporal spectral imagery series along with the image with the missing data to predict the missing values. For our dataset, we chose to sample a new 1000 locations globally, similar to how our base dataset was created. We then did a 60-20-20 training-validation-test split and used patch series from the 600 patches for finetuning. We ensured that there is no overlap in regions across the 1000 newly sampled patches.

### 7.3.2 ARCHITECTURE AND IMPLEMENTATION DETAILS

In our downstream task methodology, we would pass a series of input images with one or more timestamps having missing data, and we would ask our architecture to reconstruct the entire series, but take only the mean squared loss on the timestamps with missing values, using those loss values for backpropagation. Since our output is a series of images, we can use the same architecture as the pretraining model. However, we freeze the encoder and reinitialise the decoder weights to random values and update only these layers. We chose a input series length of 6, and varied the amount of

missing values in input to create many finetuned models, each of whose results are depicted in the next section. To simulate missing data, we would zero out large blocks from the image and pass an input series with the blacked out images to the model. For our study, we chose to blacken out 2 images per series, both with a particular percentage of missing values.

### 7.3.3 Performance on Missing Image Prediction Task

Table 2 compares the Mean Squared Errors across the various finetuned models. We can see that as the percentage of missing values increases, the MSE values go up, which can be expected due to the task getting harder. However, we can note that the VSF pretrained models

Table 2: *Comparison on downstream task of missing image imputation across models finetuned from different pretraining tasks. Mean Squared Errors using various levels of missing image percentages shown*

|  | Missing Image Prediction Finetuned Models | | | |
|---|---|---|---|---|
| % Missing | SM-MAE | MM-MAE | SM-VSF | MM-VSF |
| 50% | 792.68 | 788.94 | 362.02 | **326.43** |
| 70% | 820.46 | 814.75 | 394.32 | **337.79** |
| 90% | 826.23 | 820.43 | 404.32 | **343.88** |

have significantly lower MSE values when compared to MAE variants. We also note that MM-VSF performs the best across all models and even as missing value percentage is increased the error does not rise as much as the other variants. This is because the multimodal nature of MM-VSF helps in filling in the missing values. We also observe that the difference between SM-MAE and MM-MAE is not very high, thus furthering our hypothesis that MM-MAE method of pretraining does not effectively capture the relationship between the different modalities of data.

In Figure 6 we compare some output images from our 50 percent missing values experiment across the four methodologies. Each row corresponds to a test sample, and a common trend across rows is that finetuned models pretrained using MAE based methodologies do not fill the missing portions very well. In the first row, we can see that SM-VSF has added some false greenness but MM-VSF has not, showing that due to weather information, MM-VSF knows that growth has not occurred. Row 2 depicts a dried up river bed, but

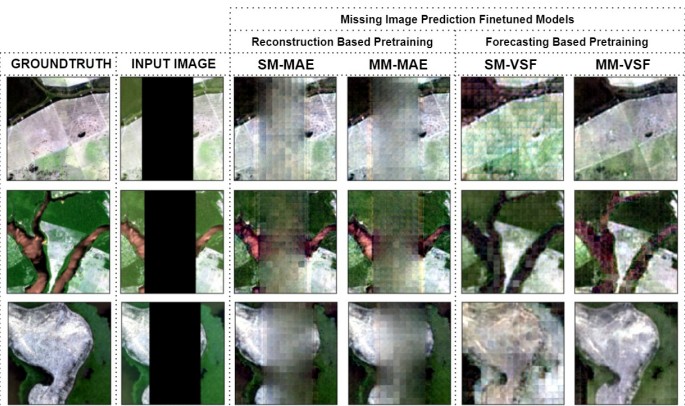

Figure 6: *Comparison of predictions for 50% missing image values across finetuned models from different pretraining tasks.*

one can observe that SM-VSF fills the image with a wet river bed, whereas MM-VSF correctly estimates the dried up bed, once again proving that weather information has helped here. In the final row, we see a case where SM-VSF did not green up the region, but MM-VSF did, which once again is a result of weather information. From these experiments, we can see that pretrained embeddings from MM-VSF are far better for finetuning for missing image imputation when compared to other methodologies of pretraining.

## 8 Conclusion

In this paper we proposed a novel multimodal spatiotmeporal foundation model, MM-VSF, that uses input of satellite and weather data and a knowledge guided pretraining task of variable step forecasting to capture the causal relationship between the two modalities. This leads to superior embeddings when compared with embeddings achieved by models using single modality input and trained with standard pretraining task of reconstruction. Our pretraining task evaluation of MM-VSF's forecasting ability showed that our foundation model is able to learn aspects that go beyond temporal autocorrelation. We showed that MM-VSF can be finetuned for a crop mapping model that is generalisable across years and also can create a good model for missing image prediction. Our model is temporally flexible and can adapt to various geoscience downstream tasks that include spatiotemporal remote-sensing data. To the best of our knowledge, our study is the first step towards incorporating knowledge guided principles in pretraining tasks and adapting multimodal approaches to improve embeddings.

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

# A APPENDIX

## A.1 PRETRAINING STAGES

Figure 7 depicts a diagrammatic representation of the 2 phase stage wise pretraining we propose to ensure that our framework's embeddings are able to capture the interaction between modalities.

## A.2 PRETRAINING RESULTS: RECONSTRUCTION

Figure 8 depicts a comparison of SM-MAE and MM-MAE on some test samples. We can observe that MM-MAE does better than SM-MAE, but only so slightly. We can notice small differences in each row (lack of greenness, blocky pixels, slight false greenness). Even on comparing Mean Squared error between the two approaches there was not a big difference.

## A.3 DOWNSTREAM TASK: CROP MAPPING

Figure 9 depicts the Cropland Data Layer(CDL) labels and the region of analysis (Sentinel Tile T11SKA) for our crop mapping downstream task. As can be seen our region of analysis lies in the heart of the California Central Valley and contains numerous crop classes. Figure 10 depicts the general layout of the architecture used for the crop mapping downstream task. The embedding series mentioned would correspond to the series $Emb_{STW}$, i.e the output series of the encoder. The decoder mentioned is a task specific decoder that use upscaling and convolution layers to map the final embedding to a pixel wise crop map.

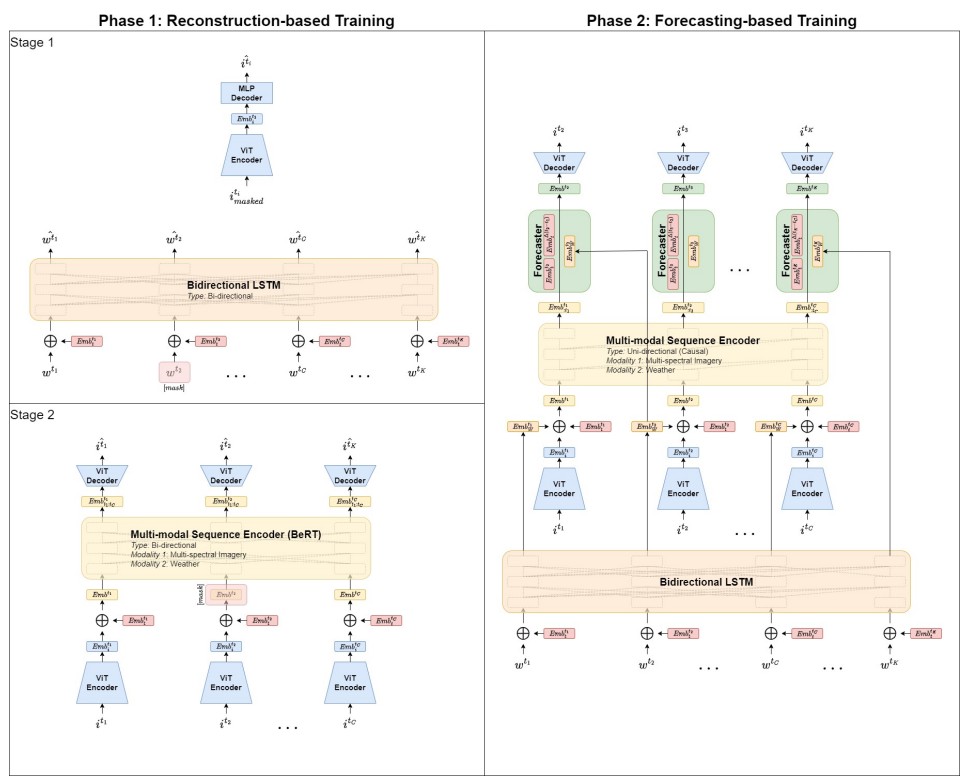

Figure 7: *A diagrammatic representation of the proposed stages of pretraining*

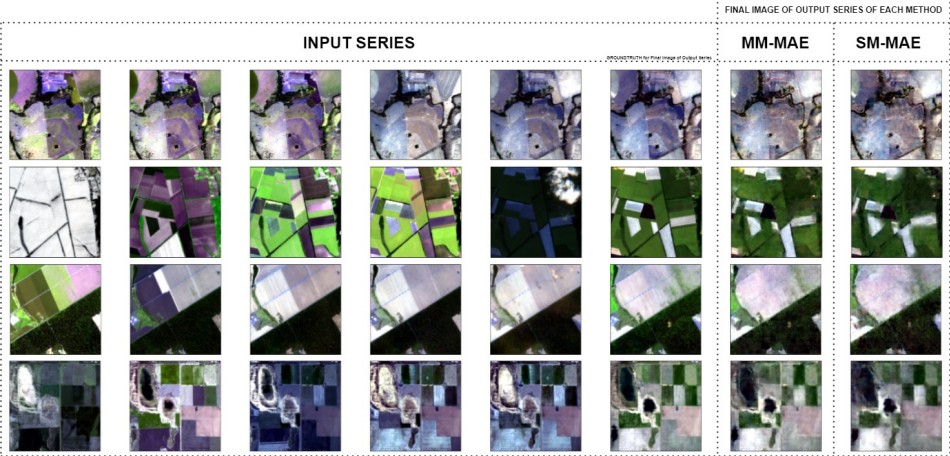

Figure 8: *A comparison of MM-MAE and SM-MAE on some test samples. As can be seen, MM-MAE is slightly better than SM-MAE.*

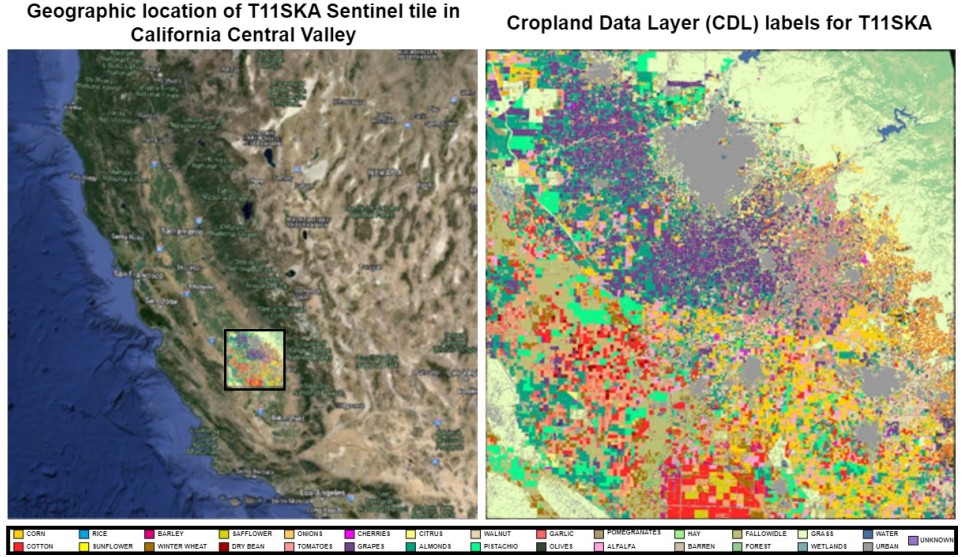

Figure 9: *Geographic location of the T11SKA Sentinel Tile and its corresponding CDL labels. Each color in the CDL image corresponds to a land cover class.*

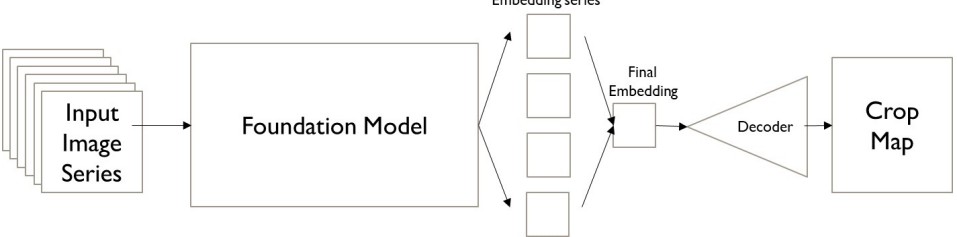

Figure 10: *Layout of architecture for downstream task of crop mapping*