# OpenReview forum: "Towards a Knowledge guided Multimodal Foundation Model for Spatio-Temporal Remote Sensing Applications"
_ICLR.cc/2025/Conference — ICLR 2025 Conference Withdrawn Submission_

### Official Review · Reviewer_DHFp · 2024-10-27

**Soundness:** 2
**Presentation:** 3
**Contribution:** 3
**Rating:** 5
**Confidence:** 3

**Summary:**

The paper presents a pretraining framework called MultiModal Variable Step Forecasting (MM-VSF). The proposed foundation model leverages multimodal data, specifically satellite imagery and weather data, to improve spatio-temporal remote sensing tasks like crop mapping and missing image prediction. The core idea is to pretrain the model using a forecasting task that captures causal relationships between the modalities, improving generalization and representation quality for downstream tasks.

**Strengths:**

The introduction of a multimodal pretraining task adds a novel aspect to the existing methods in the geoscience foundation model landscape. The paper is generally well-structured. The ability to generalize across years and handle missing data offers practical utility in real-world settings.

**Weaknesses:**

1. Insufficient experiments: My main concern about this paper is the lack of comprehensive experiments. The baseline models could be expanded, for example, by including comparisons with contrastive-learning-based methods to provide a more holistic evaluation.

2. Lack of fine-grained ablation studies: The paper could benefit from more fine-grained ablation studies, such as experimenting with different masking ratios and strategies. This would help clarify the specific impact of these hyperparameters on model performance.

3. Limited downstream tasks: The range of downstream tasks is insufficient. Including tasks like urban semantic segmentation mapping, which is highly relevant in the field of remote sensing, would strengthen the generalizability of the approach.

4. Incomplete related work: The related work section is not sufficiently comprehensive and clear. There is a lack of discussion and comparison with relevant multimodal foundation models in remote sensing, particularly models that also utilize satellite and weather data [1-2]. Incorporating these references and analyzing their similarities and differences with the proposed approach would offer better context and positioning.

5. Figures: Figures 1 and 2 have unclear text, which may hinder the understanding of key concepts. Improving the clarity of the text in these figures would enhance the overall presentation quality.

[1] Ravirathinam, Praveen, et al. "Combining Satellite and Weather Data for Crop Type Mapping: An Inverse Modelling Approach." Proceedings of the 2024 SIAM International Conference on Data Mining (SDM). Society for Industrial and Applied Mathematics, 2024.
[2] Nedungadi, Vishal, et al. "MMEarth: Exploring multi-modal pretext tasks for geospatial representation learning." arXiv preprint arXiv:2405.02771 (2024).

**Questions:**

1. How does the variable step forecasting task handle significant time gaps in satellite data? For example, how does it manage situations where satellite imagery is missing for extended periods?
2. Why the shape of weather data is 1x1？
3. In the crop mapping task, how sensitive is the performance of MM-VSF to the temporal resolution of the input data? Would increasing or decreasing the frequency of the input images change the results significantly?

---

### Official Review · Reviewer_oCWi · 2024-10-27

**Soundness:** 2
**Presentation:** 1
**Contribution:** 2
**Rating:** 3
**Confidence:** 4

**Summary:**

This paper proposes an innovative foundational model framework for geoscience, which undertakes the task of forecasting future spectral imagery using weather information and spectral imagery from previous timestamps. Through the construction of a MultiModal Variable Step Forecasting (MM-VSF) paradigm, the authors illustrate the superiority of MM-VSF in two downstream tasks: crop mapping and missing image prediction. When compared to MAE pre-training and single-modality scenarios, the proposed method demonstrates enhanced performance, thereby validating the significance of incorporating weather information in certain remote sensing tasks.

**Strengths:**

1. The concept of establishing a geoscience foundational model from the perspective of forecasting future spectral imagery is novel, and the utilization of weather information in this context has not been fully explored by the community before. This paper presents an intriguing solution tailored specifically to a foundational model for geoscience.
2. Overall, the writing is satisfactory; however, there are some significant drawbacks that I will address later.

**Weaknesses:**

1. The fundamental assumption may be flawed. While it is reasonable to assert that weather can be an influential factor in certain land use or land cover predictions, forecasting spectral imagery solely based on weather and previous spectral imagery clearly overlooks essential factors, such as human activities. For instance, in crop mapping predictions, farmers may choose not to cultivate any crops in a given year, a decision that is entirely independent of weather conditions. Similarly, the segmentation of urban functional areas is also not contingent upon weather; the construction of buildings occurs regardless of weather considerations. Therefore, the current pre-training task appears to be an inadequate choice for addressing a wide range of remote sensing tasks, rendering the fundamental assumption of this paper unconvincing.
2. The presentation of the main methodology lacks clarity. For instance, the data format for the weather information is ambiguous, and it is unclear how the bi-directional LSTM processes this weather data. I suggest providing a formal description of both the input and output data to illustrate the data flow pipeline, as this would offer a clearer overview of the methodology. Furthermore, the rationale behind selecting ViT instead of SwinViT, as well as the choice of LSTM over GRU, is not explicitly addressed. It appears that the selection of these building blocks is somewhat arbitrary and lacks specific consideration within the framework's design.
3. The experimental results presented are primarily an ablation study. In the main experimental section, the authors compare the proposed MultiModal Variable Step Forecasting (MM-VSF) model solely with various combinations of single modalities and MAE-based pre-training methods. The current results demonstrate only the necessity of the weather modality and the superiority of the Variable Step Forecasting over MAE. However, there is insufficient evidence to support the assertion that MM-VSF is a suitable choice for universal remote sensing interpretation tasks. I strongly recommend that the authors compare MM-VSF with well-established remote sensing foundational models, such as SkySense (CVPR 2024), SatMAE++ (CVPR 2024), and DeCUR (ECCV 2024), using recognized benchmarks such as BigEarthNet, fMoW, and DIOR. Additionally, the potential of contrastive learning as an avenue for developing remote sensing foundational models is not adequately addressed in this paper.
4. The analysis of the learned causal relationships is notably absent. In the abstract and introduction, the learned causal relationship between weather and spectral imagery is presented as a significant contribution of this paper; however, a detailed analysis appears to be lacking. I would appreciate the inclusion of quantitative results or mathematical induction, as causality is a well-established mathematical concept, to further elucidate this aspect.
5. Several important related works appear to be missing from the discussion. Regarding weather forecasting, the authors should consider including GraphCast (Science 2023) and Fuxi (Nature Communications 2024). Additionally, in the realm of contrastive learning, notable contributions such as DeCUR (ECCV 2024) and DINO-MM (IGRASS 2022) should also be acknowledged.

**Questions:**

1. In light of Weakness 1, the authors are encouraged to clarify the motivation and considerations underlying the proposed method.
2. Could you provide a detailed explanation of the design rationale behind your framework?
3. CDL is widely recognized for its association with noisy labels. Do you implement any label correction on this data, and if so, what specific steps do you undertake?

Additionally, the authors are encouraged to address the concerns outlined in the Weakness section.

---

### Official Review · Reviewer_bMkP · 2024-10-28

**Soundness:** 3
**Presentation:** 2
**Contribution:** 3
**Rating:** 5
**Confidence:** 3

**Summary:**

This paper proposes a foundational framework for exploring the causal relationships between temporal textual data and remote sensing imagery. The authors first use a Bidirectional LSTM to model the temporal text, followed by a Unidirectional Transformer to extract the causal relationships between the textual and image features. Ultimately, this framework can generate corresponding remote sensing images based on weather data.

**Strengths:**

The authors applied variable-step prediction, commonly used in time series forecasting, to the pretraining task of a remote sensing multimodal foundational model and achieved performance improvements in two downstream tasks. This is a preliminary exploration of a remote sensing multimodal foundational model in spatiotemporal prediction.

**Weaknesses:**

1. The impact of weather on the environment is a highlight of this paper and should be its most compelling section. However, this topic is only briefly mentioned in the abstract and introduction. Unfortunately, no subsequent experiments are conducted to discuss the specific effects of weather on remote sensing imagery, which is quite regrettable. The explanation of the causal relationship between weather and imagery seems to be solely connected to the use of the causal Transformer model, which feels somewhat tenuous. I suggest that the authors present several cases to demonstrate how different weather models affect the prediction outcomes or conduct an ablation study to isolate the contribution of weather data to the model's performance.

2. The model's performance is limited. Specifically, in Figure 8, I even feel that the MM-MAE results visually appear superior to those of SM-MAE. The SM-MAE results seem somewhat more blurred. Additionally, the images in Figures 6 and 8 are quite similar. I would encourage the authors to provide more qualitative analysis to demonstrate their method's advantages better. I suggest that the authors conduct a detailed analysis of challenging individual cases, or visualize the learned representations to illustrate how they capture weather-related information.

3. The entire paper seems to rely on only two quantitative analysis experiments, presented in Tables 1 and 2, which feel somewhat insufficient. The authors have designed a complex encoder composed of multiple components, but no ablation studies have been conducted to justify the selection of each component. This makes it difficult to be convinced that the author's choices are optimal. I suggest the authors include ablation experiments regarding the model architecture. For example, the additional experiments mentioned above.

**Questions:**

1. Why is causal encoding applied to the fused multimodal embeddings, while Bi-LSTM is used for the weather data? Logically, weather data should also be unidirectional.

2. In Figure 6, the non-restored areas in the SM-VSF results exhibit mosaic artifacts and appear inconsistent with the original image. Does this method affect regions of the image outside the restored areas?

---

### Official Review · Reviewer_P6P8 · 2024-11-03

**Soundness:** 1
**Presentation:** 1
**Contribution:** 1
**Rating:** 1
**Confidence:** 4

**Summary:**

The paper presents an approach that aims to integrate physics-driven weather data with observation-based remote sensing data to investigate the correlation between these two modalities.

**Strengths:**

The topic is interesting and valuable. The authors commendably attempt to combine physics-driven weather data with observation-based remote sensing data to explore the correlation between these two modalities.

**Weaknesses:**

1. Some opinions and assertions in the paper lacks experimental or theoretical support. For example:
    * Line093: The claim that "temporal flexible architecture is useful for generalizing across downstream tasks" is presented without the necessary theoretical backing or experimental validation.
    * Line096: The assertion that the embedding created by the proposed architecture is richer than those using reconstruction tasks is only substantiated by results from the authors' framework, lacking comparison with other methods and relevant theoretical derivation.
2. Unclear Methodology and training detail description.
    * The description of the methodology lacks formalization, rendering the training process unclear.
    * There is insufficient detail regarding the network architecture and training procedures such as the hyper-parameters, training and fine-tuning epoch number, batchsize, and etc.
    * The language employed in the paper lacks academic rigor, featuring informal expressions (e.g., using "let us look at …”), grammatical errors, improper punctuation,  formatting issues (e.g., using $\times$ instead of x), incorrect capitalization, and various typos. A revision of the writing is strongly recommended.
3. Deficiencies in Experimental Design.
    * The baseline comparisons in the experimental part can only be classified as ablation studies, as they do not include comparisons with other foundational models.
    * In the downstream task validation, comparisons are limited to the authors' ablation models, without benchmarking against state-of-the-art models in the relevant application field.
    * Given that many pretrained ViT models (even without fine-tuning in the RS domain) achieve commendable performance in remote sensing tasks, the authors should also compare their results with those of pretrained models.
    * There is a lack of experimental design concerning the model's generalization capabilities. As a foundational model, the experiments are supposed to involve more types of downstream tasks and cover remote sensing data from more satellites.
4. Suboptimal Experimental Results. The visual results in this paper appear to be of limited utility, and there is also a lack of comparison with results from other MAE-based or contrastive learning-based foundation models.

**Questions:**

Both the statements and technical aspects should be refined for rigor. A thorough revision is recommended before resubmission.

---

### Note · Authors · 2024-11-27

**Comment:**

Thank you for all the comments!

**Withdrawal Confirmation:**

I have read and agree with the venue's withdrawal policy on behalf of myself and my co-authors.